# Reproducibility Study of "Languange-Image COnsistency"

**Patrik Bartak**[†]                                              *patrik.bartak@student.uva.nl*
*Informatics Institute, University of Amsterdam*

**Konrad Szewczyk**[†]                                        *konrad.szewczyk@student.uva.nl*
*Informatics Institute, University of Amsterdam*

**Mikhail Vlasenko**[†]                                        *mikhail.vlasenko@student.uva.nl*
*Informatics Institute, University of Amsterdam*

**Fanmin Shi**                                                      *fanmin.shi@student.uva.nl*
*Informatics Institute, University of Amsterdam*

[†] *These authors contributed equally to this work.*

**Reviewed on OpenReview:** *https://openreview.net/forum?id=FvxTseSYRk*

## Abstract

This report aims to verify the findings and expand upon the evaluation and training methods from the paper LICO: Explainable Models with Language-Image COnsistency. The main claims from the original paper are that LICO (i) enhances interpretability by producing more explainable saliency maps in conjunction with a post-hoc explainability method and (ii) improves image classification performance without computational overhead during inference. We have reproduced the key experiments conducted by Lei et al.; however, the obtained results do not support the original claims. Additionally, we identify a limitation in the paper's evaluation method, which favors non-robust models, and propose robust experimental setups for more comprehensive quantitative analysis. Furthermore, we undertake additional studies on LICO's training methodology to enhance its interpretability. Our code is available at `https://github.com/konradszewczyk/lico-reproduction`.

## 1 Introduction

Despite deep neural networks showing state-of-the-art performance in numerous computer vision tasks (Chai et al., 2021), their application in safety-critical tasks like medical diagnosis is limited due to their black-box nature, making it hard to understand how decisions are made. While contemporary *post-hoc explanation* methods like Grad-CAM (Selvaraju et al., 2019) or RISE (Petsiuk et al., 2018) can increase the transparency in these systems, they cannot be used to improve the decision-making process underneath to make it more consistent with human intuition.

Previous *model-based* methods proposed improving the models by enforcing consistency constraints on the post-hoc explanations generated during training (Pillai et al., 2022; Pillai & Pirsiavash, 2021). In the reviewed study, Lei et al. (2023) propose *Language-Image COnsistency* (LICO), a training framework for enhancing model interpretability. Unlike the previous methods, LICO does not utilize post-hoc explanations during training. Instead, the loss enforces consistency between the manifold of visual features and the manifold of class-aware semantic information from a text encoder of a pretrained *vision-language model* (VLM), such as CLIP (Radford et al., 2021). By doing so, the authors claim to achieve both improved interpretability and classification performance.

In recent years, there has been a growing concern about the reproducibility of AI research (Baker, 2016; Hernández & Colom, 2023), where it is becoming increasingly difficult to reproduce and validate the findings of the papers, which undermines trust in the scientific community and leads to the waste of resources spent on pursuing unreproducible results. The causes of this crisis include incomplete documentation of the experimental procedures or introduced methods (Semmelrock et al., 2023) as well as the need for significant computational resources (Nature Computational Science Editorial Board, 2021). For this reason, we attempt to verify whether the LICO authors' claims can be reliably reproduced based on the paper. Our analysis is extended with additional experiments evaluating the interpretability of the trained models and modifications to LICO. The following are the main contributions:

- **Reproducibility Study:** We reproduce the key experiments conducted by Lei et al. to find out which of their claims can be reliably reproduced by following the paper. Furthermore, we provide information about the computational costs required during the reproduction.

- **Improved Codebase:** The code delivered by authors is incomplete and cannot be executed without non-trivial work. We provide our re-implementation and documentation of the LICO method, together with the experimental setup demonstrating its interpretability and classification performance.

- **Extended Evaluation:** We provide justification that the Insertion and Deletion tests used by Lei et al. are not indicative of the model's interpretability. Instead, we propose an evaluation setup for qualitative analysis of alignment between saliency maps and human expectations (*priors*).

- **LICO Extensions:** We conduct additional experiments on the language knowledge extraction component of LICO by (i) introducing trainable prompts specific to each class and (ii) positioning the class label tokens before the trainable prompts with the goal of improving the method.

Our work is structured as follows. Section 2 presents the claims of the paper we attempt to replicate. In section 3, we introduce the method proposed by Lei et al. and describe the experimental setup used. Then, section 4 covers the results of both replicated and extended evaluation conducted by us. The report concludes with section 5, where we discuss our findings about the reproducibility of LICO.

## 2 Scope of Reproducibility

A central idea in *model-based* methods comparable with LICO is enforcing consistency with an additional human prior during training (Han et al., 2021; Pillai & Pirsiavash, 2021; Pillai et al., 2022). In LICO, this prior takes the form of semantic information encoded within a pretrained text encoder, which should allow for training more interpretable models without compromising the quality of classification. In this work, we will focus on the following claims by Lei et al.:

1. **Enhanced Interpretability:** By matching the manifolds of visual and text embeddings, the semantic information from the pretrained text encoder can guide the image classification model to focus on distinguishing features of objects. As a result, the explanations for model decisions are more consistent with human expectations, which is reflected in both quantitative and qualitative analyses.

2. **Improved Classification Performance:** As opposed to the previous methods increasing the interpretability of image models, LICO does not bring a deterioration to the classification performance and, in some cases, may even result in increased accuracy, when compared with baseline models trained without LICO.

3. **Necessity of both Manifold Matching and Optimal Transport Loss:** LICO introduces a training method that supplements the traditional cross-entropy loss with two additional loss components (described in section 3.1), both with a goal of bringing the manifolds of text and visual features closer to each other. Although these two components serve similar functions, the mutual presence of both is beneficial to the quality of the resulting models.

In addition to validating the claims brought in the original paper, we analyze the impact of the LICO method on the training times of the model. Moreover, we extend the interpretability study by proposing an enhanced experimental setup consisting of three quantitative tests.

## 3 Methodology

This section describes the approach taken to reproduce the work. We start with an overview of the LICO algorithm, and then continue by describing the datasets and hyperparameters used. We then discuss the experimental setup and code - specifically, which parts of the authors' implementation were missing or incomplete, the parts we re-implemented based on the paper, and the resources that we used. Lastly, we introduce the quantitative metrics that we use for the evaluation.

### 3.1 The LICO Algorithm

LICO is a training framework for supervised image classification models that utilises a pre-trained text encoder (e.g. from CLIP models) to guide the training process of the image classifier such as ResNet-18. This is achieved by supplementing the the standard Cross-Entropy (CE) loss used to train the model by adding Manifold Matching (MM) and Optimal Transport (OT) loss components, weighted by $\alpha$ and $\beta$ parameters respectively. A general description of both is provided in the following paragraphs.

The idea behind LICO is presented in fig. 1 with the most important concepts to consider being image manifold and language prompt manifold. The image manifold is defined by feature maps $f_\theta(x_i)$ from the produced by CNN within the image classifier, and the language prompt manifold is defined by projected text embeddings $h_\psi(g_\phi(t_i))$ obtained by passing a prompt through a pretrained Vision-Language Model (VLM) and a trainable MLP (which matches the dimensionality of the visual feature maps and the text embeddings). This prompt consists of the ground truth class label and several trainable tokens.

The MM and OT losses aim to align these two manifolds, such that the image manifold becomes enhanced with the semantic structure allegedly present in the language prompt manifold due to the multi-modal training of the VLM from which the embeddings are extracted. The goal of MM loss is to provide coarse alignments between the two manifolds based on the similarities between pairs of samples within each mini-batch, while OT loss is responsible for establishing a correlation between text tokens and feature maps for individual data points.

Other than images and their ground truth class labels, no additional information is needed to make use of LICO. After training, everything except the image model is discarded. This means that a model trained using LICO only differs from a non-LICO model by its parameters, and LICO therefore does not increase the computational cost at inference.

### 3.2 Datasets

We train and evaluate the presented models on two image classification datasets. Following the original experiments, we use **CIFAR-100** (Krizhevsky et al., 2009), which provides 50000 training and 10000 validation images divided into 100 classes. Additionally, we use **ImageNet-S**$_{50}$ (Gao et al., 2022), consisting of 64431 training images and 752 validation images with segmentation masks and bounding box information that we use for extended evaluation. Both datasets are balanced, containing approximately the same amount of samples of each class. The preprocessing performed during training is described in appendix B.

The selection of these datasets, and not full ImageNet-1k, has been dictated mostly by the computational limitations we have encountered during our work. However, we believe those to be sufficient to show whether the main claims of the LICO paper hold. As the original authors have used CIFAR-100 in their work, we should be able to see the same trends present in our results as they have. However, we acknowledge the fact that limited resolution of images within CIFAR-100 may limit the ability of the method to connect the visual cues with semantic information from VLM. For this reason, we have chosen to conduct additional experiments using ImageNet-S$_{50}$ dataset, a subset of the original ImageNet-1k limited to 50 classes. The higher resolution with images remove the possibility of LICO's impact being limited by the lack of distinguishable features in

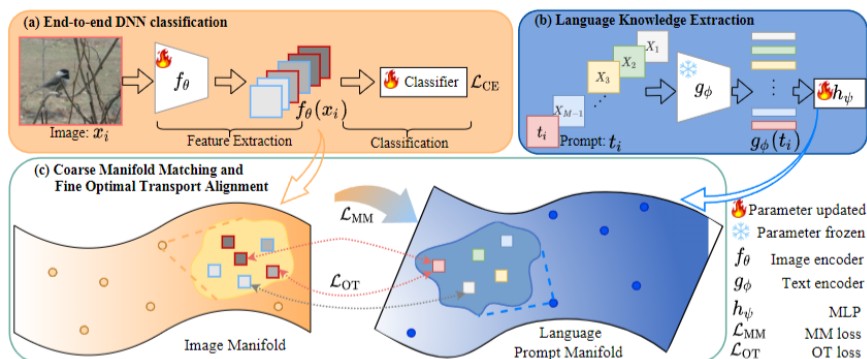

Figure 1: Framework of the LICO method. (a) Conventional classification pipeline of DNNs. (b) Language feature extraction with pre-trained text encoder. (c) Manifold matching among samples and optimal transport alignment between feature maps and prompt tokens within each sample. Figure and caption adapted from (Lei et al., 2023).

blurry images of CIFAR-100. An additional benefit of this datasets is the presence of segmentation masks and bounding boxes for objects, which we utilise in our extended analysis in the Section 4.2.1.

### 3.3 Hyperparameters

We use the original values for the hyperparameters that were specified by Lei et al. (2023): SGD optimizer with learning rate = 0.03, momentum = 0.9, weight decay = 0.0001, and cosine rate decay schedule i.e. $\eta = \eta_0 \cos\left(\frac{7\pi k}{16K}\right)$, where $\eta_0$ denotes the initial learning rate, $k$ is the index of training step, and $K$ is the total amount of training steps. The LICO-specific parameters are also used unchanged: $\alpha = 10$, $\beta = 1$, and the hidden dimension of the text projection MLP is 512. We use 100 epochs for all tested datasets and the ResNet-18 architecture unless otherwise stated.

### 3.4 Experimental Setup and Code

Some code for LICO has been made publicly available by the authors via GitHub[1]. However, as of the time of writing, the published codebase does not include a significant portion of the experimental and method-related code. While the currently available README file provides some guidance, additional details would greatly decrease the effort required for an accurate reproduction of the original experiments. Implementations of explainability methods, such as GradCam and the metrics used for evaluation, like insertion and deletion score, were not provided in the available codebase. Because of that, we use the CGC GitHub repository[2] for implementations of GradCam, insertion score, and deletion score.

Consequently, we re-implemented most of the method based on the paper, except for computing the OT loss, which is present in the authors' codebase. In our work, we follow the paper as closely as possible. Our codebase includes the complete LICO method, scripts for experiments, thorough documentation, and instructions how to reproduce the results of our extended evaluation. To reduce the amount of code needed for the implementation, and to increase readability, we use the PyTorch Lightning framework (Falcon & The PyTorch Lightning team, 2019).

Our code diverges from the formulas by Lei et al. in the way the temperature parameter of the MM loss is implemented. Inspired by the implementation of CLIP (Radford et al., 2021), we multiply by the exponent of the trainable value instead. The range of valid values in this case is the same, but the multiplicative factor is naturally in the interval $(0, \infty)$, avoiding a possible division by zero or negative temperature. Further in line with CLIP, we bound the trainable parameter to $[0, \log(100)]$. Not doing so allows a trivial solution for the MM loss: minimize the temperature to $-\infty$ and achieve a feature-independent uniform softmax output.

---

[1]https://github.com/ymLeiFDU/LICO
[2]https://github.com/UCDvision/CGC

Furthermore, the division by a unconstrained trainable parameter risks introducing the division by zero or a negative number. These potential issues were not explicitly discussed by the authors, and we addressed them by multiplying by a logarithm.

Moreover, the original paper does not provide specific details about the normalization methods applied to text and visual features in the calculation of MM and OT losses. Our initial experiments indicated that the training process with unnormalized feature vectors led to instability due to the high magnitudes of gradients. Subsequently, we observed that the magnitude of the OT loss component scales with the L2-norm of the feature vectors. To compensate for that, we conduct L2-normalisation of text and visual features before the calculation of MM and OT losses, which increases the stability of the training.

In line with the authors, we use top-1 and top-5 accuracy metrics to assess the classification performance. We also use Insertion and Deletion scores (Petsiuk et al., 2018) for quantitative analysis of model interpretability. However, in section 3.4.1, we suggest that this metric may not be the most appropriate for measuring the quality of explanations in *model-based* methods, and we propose an alternative evaluation setup.

### 3.4.1 Quantitative Analysis of Salience Maps

First, we show that the metrics used in the original paper, `insertion` and `deletion` scores, are applied in a context for which they were not originally intended. Then, we propose an extended experimental setup to quantitatively assess the interpretability of LICO and analyze the consistency of its explanations with the human prior. Our approach involves the use of three metrics: *Salience Equivariance Similarity* (SES), *Segmentation Content Heatmap* (SCH), and the *Multi Object Salience Uniformity* (MOSU).

**Insertion and deletion:** Initially proposed by Petsiuk et al., these metrics were introduced for the assessment of post-hoc interpretablity methods that produce saliency maps, not the models. The metrics work as follows: given a picture, a saliency map for the target class is extracted from the model. Then, for `insertion`, the blurred version of the image is taken, and the parts that have high saliency values are gradually unblurred. At every step, the output probability of the model for the target class is recorded. The value of the `insertion` metric is the area under the curve (AUC) of the graph with *share unblurred* (the percentage of pixels within the image being unblurred in the order of decreasing salience) as the x-axis and *output probability* of the target class as the y-axis. The `deletion` acts similarly, but instead, the evaluation starts with a fully complete image, and the high-saliency regions are greyed out. The `insertion` needs to be maximized, while `deletion` - minimized.

As such, a good metric score for a combination of a model and an explanation method merely demonstrates that the explanation method has done well. As demonstrated in appendix A, even a model that focuses on small regions and lacks robustness can perform well on these metrics, highlighting their limitations in this context.: removing a tiny part completely confuses the model, and adding that part immediately makes the model predict the correct class.

The LICO method aims to change the model itself in order to make it more interpretable. Therefore, the comparison between the `insertion` and `deletion` scores of a baseline model and a model trained with LICO is not sufficient to claim that one model is more interpretable than the other.

**Salience Equivariance Similarity:** One intuitively desirable property of saliency maps is for them to be near-equivariant under certain transformations. This idea has resulted in the previously mentioned CGC method, created by Pillai et al. (2022), which uses a contrastive loss in order to encourage saliency maps that are both equivariant (under translation and cropping), and distinct from saliency maps of other images.

Because LICO does not explicitly encourage equivariance, yet claims to outperform CGC, it is of interest to investigate whether CGC retains an advantage in this respect. We compute the SES metric, used by (Pillai et al., 2022), as follows for an image $I$:

$$\text{SES}(I) = S_C(t(s(I)), s(t(I))) \tag{1}$$

With $S_C(\cdot, \cdot)$ denoting cosine similarity, $s(\cdot)$ being a post-hoc saliency map method (in our case GradCam), and $t(\cdot)$ being some spatial transformation (in our case translation and cropping). It is important to note that for spatial transformations, this metric can be trivially maximized by simply returning a uniform saliency

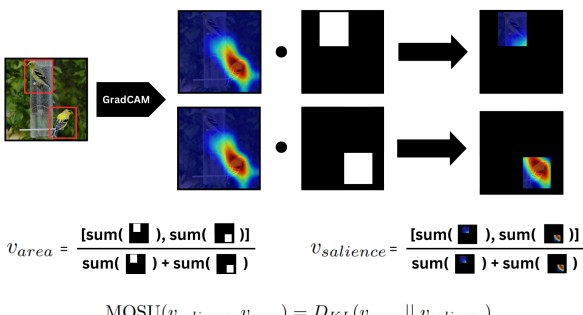

Figure 2: A graphical overview of the evaluation methods in our evaluation setup: SES, SCH and MOSU. It should be noted that for the same of clarity, the computation of MOSU metric presented in this figure is limited to two instances of objects in the images; however, all sums included in the formula are over all instance of the objects present in the image.

map for any image. It must therefore be considered in combination with other metrics, such as the one we discuss next.

**Segmentation Content Heatmap:** One of the obstacles in producing models consistent with human intuition is the reliance of image classification models on spurious features that are associated with the target object, but are not causally related to it. Hence, our second evaluation concerns the intuition that the features used by the model to classify the object ought to lie within the boundaries of the object. Using a specific variant of the Content Heatmap (CH) metric proposed by Pillai & Pirsiavash (2021), we consider a dataset where segmentation masks for the main object are provided, and take the weighted share of the saliency heatmap that lies within the segmented class. More precisely, we compute the SCH for some image as:

$$\text{SCH}(H, M) = \frac{\sum_{i,j} H_{ij} \cdot M_{ij}}{\sum_{i,j} H_{ij}} \tag{2}$$

Where $H$ is a matrix with salience values $H_{ij} \in [0, 1]$ for each pixel, and $M$ is a matrix representing a binary segmentation mask obtained from the dataset, with $M_{ij} \in \{0, 1\}$ being 1 if the pixel belongs to the target object, and 0 otherwise. Such a metric shows how much importance the model assigns to the target object. It does not penalize if only part of the object is required to perform the classification, but it discourages attending to features that do not identify the class in a way consistent with human expectations.

**Multi Object Salience Uniformity:** The last component of the proposed experimental setup provides a quantitative evaluation of the performance of the model when multiple objects of the same class are visible in the image. In such a case, a human would expect the explanation to cover all of the instances. However, the model could easily make the correct prediction based only on a single occurrence of an object. We propose to consider the distribution of salience within the bounding boxes of instances of a given class for data points where multiple instances are present. We would expect it to be proportional to the area of a bounding box encompassing the particular instance. To measure the distance between distributions, MOSU uses Kullback–Leibler divergence:

$$\text{MOSU}(v_{area}, v_{salience}) = D_{KL}(v_{area} \mid\mid v_{salience}) \tag{3}$$

Where $v_{salience} \in \mathbb{R}^n$ denotes the probability vector corresponding to the distribution of salience lying within the bounding box encompassing each of the $n$ instances of the class in an image with its elements equal to $v_{salience}[i] = \frac{\text{SalienceInBoundingBox}[i]}{\sum \text{SalienceInBoundingBox}}$ with SalienceInBoundingBox[i] being a sum of salience within the bounding box of the object $i$. In a similar manner, $v_{area} \in \mathbb{R}^n$ can be defined as a vector such that $v_{area}[i] = \frac{\text{BoundingBoxAreas}[i]}{\sum \text{BoundingBoxAreas}}$ where $i$ corresponds to each of object instances in the image. To our knowledge, MOSU is a novel metric. Since this measure does not take into account salience attributed to regions outside of objects, it should be used in tandem with the *SCH* metric or its variant utilizing bounding box information (Pillai et al., 2022).

## 3.5 Computational Requirements

Aiming to provide a more robust evaluation, we decided to run the experiments with multiple seeds, rather than once like the original authors do. To accommodate that, we needed to focus on datasets smaller than the full ImageNet-1k due to limited computational resources. For that reason, we have decided to limit our reproduction to CIFAR-100 and ImageNet-S$_{50}$ datasets as described in 3.2. This allowed us reduce the hardware requirements of our reproduction compared to Lei et al. (2023). For the experiments, we use 2 machines with the following GPUs: NVIDIA GeForce RTX 4090 (Machine 1), and NVIDIA A100-SXM4-40GB (Machine 2).

The time taken per training epoch in different configurations is provided fully in table 1. We generally observe that the LICO and CGC take significantly more time and memory to train. We also notice that the time increase depends on the model size and batch size. The table does not provide standard deviations as they are all near 0.

Table 1: Seconds per epoch for (ResNet-18/ResNet-50). Evaluated on Machine 1 at 16-bit precision.

| Training method | CIFAR100, batch 64 | ImageNet-S-50, batch 64 | ImageNet-S-50, batch 128 |
|---|---|---|---|
| Baseline | 23/68 | 42/87 | 43/91 |
| LICO | 98/173 | 127/223 | 102/-[3] |
| CGC | 107/245 | 133/316 | 135/-[3] |

# 4 Results

## 4.1 Results reproducing original paper

**Claim 1. Enhanced Interpretability**

**Insertion and Deletion Tests**: The experiments in Table 2 based on Insertion and Deletion metrics used by Lei et al. show a deterioration over the baseline, which is not in line with the authors's claims. However, due to reasons described in section section 3.4.1 this may not be sufficient evidence to judge the interpretability of models; hence, the extended quantitative evaluation is conducted in section 4.2.1.

---

[3]24 GB of VRAM is not enough to run this experiment at fp16 precision. Same experiment with the baseline takes 6.9 GB of VRAM.

Table 2: Insertion and deletion on ImageNet-$S_{50}$ using ResNet-18. "$\pm$" indicates standard deviation.

| Method | Insertion ($\uparrow$) | Deletion ($\downarrow$) | Combined = Insertion $-$ Deletion ($\uparrow$) |
|---|---|---|---|
| CE loss (baseline) | $\mathbf{.727} \pm .003$ | $\mathbf{.243} \pm .006$ | $\mathbf{.484} \pm .008$ |
| + MM loss | $.725 \pm .003$ | $.248 \pm .011$ | $.477 \pm .011$ |
| + OT loss | $.707 \pm .004$ | $.258 \pm .004$ | $.447 \pm .008$ |
| + MM & OT (LICO) | $.706 \pm .012$ | $.263 \pm .008$ | $.443 \pm .019$ |

**Visual Analysis of Saliency Maps**: To qualitatively investigate the interpretability of LICO, saliency maps were also analyzed visually. Several images were sampled randomly from the CIFAR-100 and ImageNet-$S_{50}$ datasets. The samples can be seen in fig. 3. Ideally, we would like to see that GradCam saliency maps of LICO models are less often focused on spurious features. However, the opposite can generally be observed, indicating that the saliency maps tend to be less interpretable. We also noted that saliency concentrates on the extreme edges of images for LICO models notably more often than for the baseline. This can be seen in fig. 3 for classes "snail", "bottle", and "kuvasz".

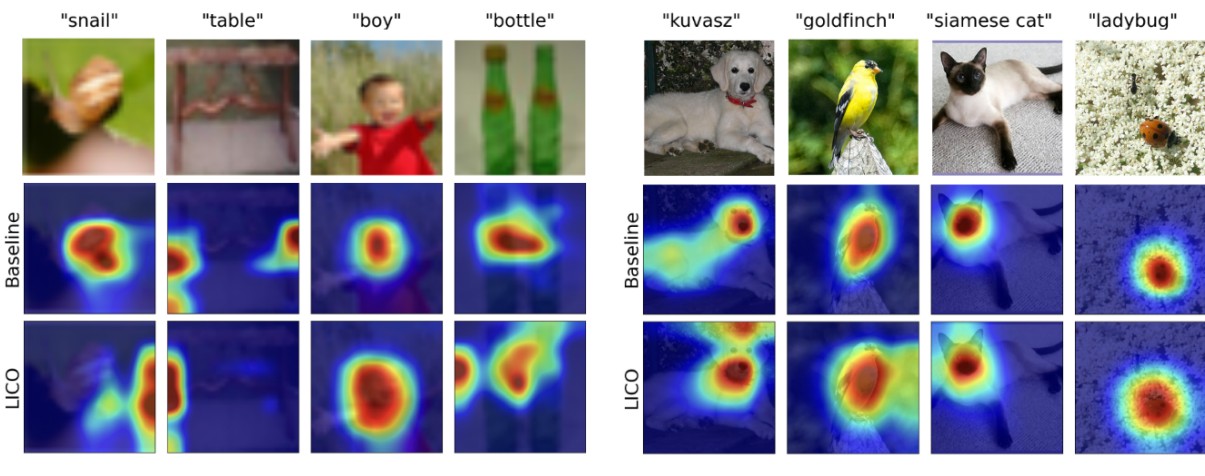

(a) Examples of saliency maps on CIFAR-100      (b) Examples of saliency maps on ImageNet-$S_{50}$

Figure 3: Examples of GradCam saliency maps for baseline ResNet-18 and ResNet-18 + LICO sampled from CIFAR-100 and ImageNet-$S_{50}$ as part of the qualitative evaluation.

## Claim 2. Improved Classification Performance

Although LICO trained on CIFAR-100 outperforms CGC model on top-1 and top-5 accuracy, it does not improve upon a baseline ResNet-18 as seen in Table 3. Likewise, the experiments conducted on ImageNet-$S_{50}$ dataset in Table 4 do not support the claim that the classification accuracy of baseline models can be improved by training with LICO.

Table 3: Classification accuracy and SES on CIFAR-100 using ResNet-18

| Method | Top-1 | Top-5 | SES |
|---|---|---|---|
| ResNet-18 | **.7431** | **.9320** | .447 |
| + CGC | .7250 | .9153 | **.949** |
| + LICO | .7309 | .9285 | .485 |

## Claim 3. Necessity of both MM and OT Loss

**Interpretability**: Experiments on ImageNet-$S_{50}$ using Insertion and Deletion metrics in Table 2 show that training with MM loss achieves a Combined score that is slightly worse compared to a ResNet-18 baseline, whereas OT loss brings significant deterioration to this metric. Combining the two losses results in a further

decrease and suggests the lack of positive interaction between MM and OT losses, contrary to the claims of Lei et al.. This will be further investigated in section 4.2.1.

**Classification Performance**: Table 4 shows that full LICO obtains lower top-1 accuracy on ImageNet-$S_{50}$ compared to model trained using only OT or only MM loss. It should be noted that while in terms of top-5 accuracy LICO slightly outperforms a model trained with only MM loss or only OT loss, it is still worse than the baseline. Overall, these results do not support the claims of LICO authors, who argue that using MM and OT losses jointly is beneficial to the result.

### 4.2 Results Beyond the Original Paper

#### 4.2.1 Extended Evaluation Setup for Interpretability

For the extended quantitative analysis of LICO's interpretability, we follow an experimental setup that has been described in detail in section 3.4.1.

**Salience Equivariance Similarity**: LICO trained on CIFAR-100 shows a small improvement over the baseline in terms of SES metric as seen in Table 3. It should be noted that CGC (Pillai et al., 2022) vastly outperforms both LICO and the baseline, which should be expected given that a similar objective is used during its training.

During the experiments on ImageNet-$S_{50}$, the full LICO achieves lower SES than the baseline as shown in Table 4. Since the SES metric measures the robustness of post-hoc explanations produced for the model to different image transforms, this suggests that saliency maps for LICO models are less stable given basic geometric transformation of the input compared to the baseline ResNet-18 model. However, it should be noted that models trained using only one of the OT and MM losses slightly outperform the baseline, which shows the negative effect of combining two losses.

**Segmentation Content Heatmap**: The results for the experiments on ImageNet-$S_{50}$ in Table 4 show that the explanations generated for baseline models tend to be more localized and contained within the boundaries of the relevant objects compared to LICO models as measured by SCH. Although the models trained in the ablation study on loss components perform worse than the baseline on SCH metric, they again outperform LICO models.

**Multi Object Salience Uniformity**: The results for MOSU in Table 4 show that the baseline models are better at distributing the salience between all relevant instances of a considered class proportionally to their size compared to the LICO model. The full LICO also achieves worse results than the model trained using only MM or only OT loss.

As shown by these three metrics, the explanations produced for LICO-trained models are less consistent with human intuition compared to a ResNet-18 baseline as well as models trained using only one of the additional loss components. This further confirms that there is little support for **Claim 1** regarding improved interpretability of LICO models and **Claim 3** regarding the beneficial effect of using MM and OT losses jointly.

Table 4: Accuracy and interpretability of loss ablation on ResNet-18 with LICO on ImageNet-$S_{50}$

| Method | Accuracy | | Interpretability | | |
|---|---|---|---|---|---|
| | **Top-1** | **Top-5** | **SES** ($\uparrow$) | **SCH** ($\uparrow$) | **MOSU** ($\downarrow$) |
| CE loss (Baseline) | $.8218 \pm .0132$ | $\mathbf{.9525} \pm .0045$ | $.914 \pm .001$ | $\mathbf{.459} \pm .002$ | $\mathbf{.216} \pm .013$ |
| + MM loss | $\mathbf{.8227} \pm .0049$ | $.9468 \pm .0039$ | $.916 \pm .002$ | $.427 \pm .016$ | $.233 \pm .013$ |
| + OT loss | $.8187 \pm .0035$ | $.9494 \pm .0038$ | $\mathbf{.917} \pm .001$ | $.411 \pm .004$ | $.229 \pm .032$ |
| + MM & OT (LICO) | $.8160 \pm .0027$ | $.9503 \pm .0023$ | $.901 \pm .002$ | $.396 \pm .014$ | $.253 \pm .024$ |

#### 4.2.2 LICO Extensions for Improved Interpretability

In our attempt to reproduce the experiments, we noticed that the text features for the context tokens encoded by CLIP were the same for all classes, which limits the chances of identifying meaningful features in aligned visual feature maps. This happens due to the CLIP's text transformer encoding the text features based only

on the previous tokens by using masked self-attention (Vaswani et al., 2017). In such a setup, the context tokens cannot encode information useful to a specific class as the encoder cannot attend to class labels that are appended after them. Identifying this as a potential limitation of LICO, we propose two extensions targeted at resolving this problem.

**Class-Specific Trainable Context:** Using class-specific trainable context tokens preceding the class labels will allow contexts that only have to fit one specific class. As a result, better text-image alignment can be achieved for all classes. The experiments on ImageNet-$S_{50}$ in Table 5 show that this improves both the classification performance and interpretability of the trained models over the vanilla LICO method.

**Front Placement of Class Label:** Since the text encoder of CLIP attends only to the previous tokens, we propose changing the position of class labels to the front of the prompt. This should allow for producing text features of the context tokens that would encode class-aware semantic information. Although this leads to the deterioration in classification performance over LICO, there is a significant improvement in terms of model interpretability as seen in Table 5.

Table 5: Accuracy and Interpretability of Prompt Variants on ResNet-18 with LICO and ImageNet-$S_{50}$

| Method | Accuracy | | Interpretability | | |
|---|---|---|---|---|---|
| | **Top-1** | **Top-5** | **SES** ($\uparrow$) | **SCH** ($\uparrow$) | **MOSU** ($\downarrow$) |
| LICO | $.8160 \pm .0027$ | $.9503 \pm .0023$ | $.901 \pm .002$ | $.396 \pm .014$ | $.253 \pm .024$ |
| LICO + class-specific context | **.8191** | **.9574** | **.908** | .418 | .219 |
| LICO + class label at the front | .8045 | .9441 | .905 | **.419** | **.206** |

## 5 Discussion

In this work, we were unable to reproduce the claims made by Lei et al. (2023). We conducted several experiments, some of which replicated the original authors' evaluations, while others expanded upon the evaluation methods and the LICO model itself.

The first claim of enhanced interpretability was not reproduced. Qualitatively, produced saliency maps on ImageNet-$S_{50}$ tend to cover spurious features more often than for the baseline. For CIFAR-100, this is even more the case. In some instances, LICO resulted in artifacts appearing around the borders of the saliency maps, which were not present in maps for baseline models. Quantitatively, LICO saliency maps result in insertion, deletion, and combined scores that are not significantly better than baseline scores.

Likewise, we did not observe significant improvements in classification performance compared to the baseline as stated in the second claim we strove to reproduce. For both CIFAR-100 and ImageNet-$S_{50}$, LICO models result in lower or similar top-1 and top-5 accuracy than the baseline model.

Finally, we were also unable to confirm the third claim regarding the necessity of both MM and OT loss. Ablating individual loss components did not lead to a decrease in either classification performance or model interpretability for any of the evaluations. In most cases, it rather showed an improvement over the full LICO approach.

Our extended analysis highlighted additional areas where the performance of LICO may be limited. The saliency maps generated for LICO models appear to be less equivariant to translations and crops compared to the baseline and also appear to cover more of the regions outside the boundaries of the target objects. Moreover, the explanations for LICO are worse at highlighting all instances of the relevant class compared to the baseline model.

Furthermore, we observed improved accuracy and interpretability when incorporating class-specific trainable context and positioning class labels at the front compared to utilising original shared prompts. This suggests that more relevant and descriptive text features may lead to improvements in the quality of the aligned feature maps.

**Future work:** We believe the idea of incorporating the text features into an image classifier is an interesting way of improving model interpretability. It might be able to hint at human understanding of the image,

especially if longer descriptions are used. Thus, we propose to explore using image captions to provide more relevant signals for aligning image features.

**Limitations:** Our results may be limited by several factors. Computational constraints prevented us from conducting experiments using the ImageNet1k dataset, on which the authors claim their method performs well. That being said, LICO was also reported to perform well on CIFAR-100, and this did not turn out to be the case in our reproduction. The differences in results might have also arisen due to the potential use of regularisation or normalization methods not mentioned in the paper during the original experiments. The question regarding that was part of our attempted communication with the authors; however, we had to implement the method without their response.

## 5.1 What was Easy

From a technical perspective, most of the content of the paper was approachable. The explanations in the paper are quite clear, except for a few instances — for example, the cost function $c$ in eq. (4) not being defined. Hyperparameters were included where relevant, which made it quite straightforward to set up the experiments replicating the authors' claims. Moreover, in addition to working on ImageNet-1k, Lei et al. also conducted experiments on smaller datasets like CIFAR-100, which allows for attempts at reproduction even with a limited computational budget.

## 5.2 What was Difficult

The most time-consuming part of our reproduction study was the full re-implementation of LICO based on the paper's description, as the provided codebase was partial. At the time of writing, the code on the LICO repository contains only the implementation of Sinkhorn distance used in OT loss and the partial code for the calculation of MM loss. However, the latter deviates from the text of the paper — the main difference being the calculation of softmax on distances rather than negated distances as described in the paper. This choice was not justified by comments in the codebase. There is also no code with a complete implementation of the training loop or any of the conducted experiments.

## 5.3 Communication with Original Authors

As described in the previous sections, the official LICO code published by Lei et al. is not sufficient to reproduce the paper without reimplementing the majority of the method. Moreover, it should be noted that there was no code publicly available in the repository before the 12th of January 2024, despite the paper having been published in the autumn of 2023.

To clarify certain implementation details not fully specified in the paper or available code, we attempted to contact the authors via email and GitHub. As we did not receive a response during our study period, our implementation decisions were based on our interpretation of the published paper and referenced literature.

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

## A  Adversarial Dataset

In order to demonstrate that deletion and insertion metrics favor non-robust models, we construct the Adversarial dataset based on ImageNet-S$_{50}$. The only difference from the base dataset is that all images for one of the classes in a dataset are marked with a small red dot of fixed size at the top left corner. With this, we aim to induce non-robustness in the model, as it is likely to start using the red dot as the sole predictor of the modified class, rather than the actual object in the picture.

We train a ResNet-18 with CE loss from scratch on this dataset and observe that high importance is assigned to the location of the red dot for the class that has it. The overall deletion score is 0.129, which is comparable to models trained on ImageNet-S$_{50}$. Meanwhile, deletion specifically for the class with the red dot is 0.026.

For the deletion metric smaller is claimed to be better, so this experiment shows that the deletion metric favors non-robust models that focus on relatively small, but perhaps irrelevant, details. It thus should not be used to assess the explainability potential of an image classification model.

## B  Data Preprocessing

During our training process, we use minimal data augmentations. Namely, `RandomResizedCrop` and `RandomHorizontalFlip` for the training set, following the available code of Pillai et al. (2022) and no augmentation for the validation and testing sets. For the Adversarial dataset (see appendix A), we remove both augmentations to ensure the red dot is always visible and always at the same spot. Instead, we just resize the images to (224, 224).

## C  Ensuring Reproducibility

To ensure reproducibility, we make all of our experiments deterministic by setting the random seed. So that no cherry-picking is involved, we fix the seeds to be: 1, 2, and 3. Due to computational constraints, we trained only the most important models on ImageNet$_{50}$ dataset on all three seeds — the remaining models were trained on seed 2.

## D  Environmental Impact

The GPUs used for the training, RTX 4090 and Tesla A100, consume about 200W during training. To that, we can add another 200W for the CPU, cooling, etc. One training run of LICO on CIFAR-100 or ImageNet-S$_{50}$ takes approximately 10000 seconds in our case. During the reproduction efforts, we estimate that an equivalent of 100 full runs were performed.

Therefore, $0.4\text{kW} \cdot 10000\text{s} \cdot 100 \div 3600\text{s/h} = 111\text{kWh}$ of power was consumed. According to the online resource Nowtricity, in the Netherlands, an equivalent of 421 grams of CO2 is emitted per kWh produced. The reproduction study thus contributed an equivalent of $111 \cdot 421 = 46.7$ kilograms of CO2 — this is roughly equal to the $CO_2$-equivalent emissions of 3 livestock cows in the Netherlands within a single day (Koning et al., 2020).

