# OpenReview forum: "Reproducibility Study of "Languange-Image COnsistency""
_TMLR — Accepted by TMLR_

### Review · Reviewer_nqwA · 2024-05-07

**Summary Of Contributions:**

The authors of this paper investigate the reproducibility of the Language-Image COnsistency (LICO) training framework first introduced in the 2023 paper “LICO: Explainable Models with Language-Image Consistency” by Lei et al. In their investigation, they found that the LICO code was incomplete and largely undocumented, which prompted the authors of this paper to develop and document their own implementation of the LICO framework. When reproducing the experiments in the LICO paper, the authors found that the claims made by Lei et al. were inaccurate. Specifically, Lei et al. claimed that the LICO training framework provided enhanced model interpretability, improved classification performance, and that the combination of two loss components, the Manifold Matching (MM) and Optimal Transport (OT) loss, improved the quality of the model. The authors showed evidence suggesting that these claims may not have been as accurate as suggested by Lei et al. To address these discrepancies, the authors not only provide suggestions to improve the LICO framework but also implement these suggestions in their experiments. The authors also extend the evaluation methodology of LICO by using the Salience Equivariance Similarity and Segmentation Content Heatmap metrics. Furthermore, they present a novel metric to represent the model's ability to uniformly highlight features from multiple instances of an object, known as Multiple Object Salience Uniformity (MOSU).

**Audience:**

Yes

**Claims And Evidence:**

No

**Requested Changes:**

I know that having a good connection with the authors of LICO may be problematic, but given current situation I feel that this seems necessary to make the submission more convincing, no matter how the submission is modified later. Alternatively, if the authors of the submission can reproduce all the results in LICO, these can be considered strong evidence for the correctness of the reimplementation, on top of which, the authors can discuss more.

Also, there are some minor changes that can enhance this submission. I think that adding the original images used in figure 2. alongside the saliency maps would allow readers to more easily see where the model’s attention is being placed. Additionally, I believe there should be a “the” before “ablation study” in the “Segmentation Content Heatmap” subsection of section 4.2.1. Specifically, in the last sentence, where it states, “Although the models trained in ablation study on loss components…”.

**Strengths And Weaknesses:**

Strengths:

This paper is very well written overall. It provides context for the LICO framework and claims made by Lei et al. before discussing the experimental setup and results. The majority of the paper was also grammatically sound with good readability. Additionally, the analysis performed by the authors was thorough and well thought out.

Weaknesses:

My biggest concern of this submission is that all the reimplementation details have not been confirmed by the authors of LICO, which makes me difficult to determine if all the results in the experiments are correct and convincing. I have noticed that the authors of the submission discussed the bad communication with the authors of LICO, but this should not be the reason that I have to trust all the results and claims of the submission. Also, from the limitations in the conclusion section, it seems that the reimplementation cannot reproduce some results in LICO paper, such as on ImageNet, and the authors of the submission are not clear about some implementation details of LICO as well. These discussions make all the findings in the submission much less convincing to me.

Also, there are some minor concerns as follows: A weaker element is evident under claim 3 in section 4.1, where claims referencing tables do not align with the data presented in the table. For example, in the interpretability subsection of claim 3, it states that “Table 2 show that training with MM loss achieves a Combined score that is better compared to a ResNet-18 baseline…”. However, table 2 does not appear to support this claim since the combined loss under the “+ MM loss” row is lower than that of the baseline. Similarly, in the following subsection, it states, “It should be noted that while in terms of top-5 accuracy LICO slightly outperforms a model trained with only MM loss, it is surpassed by using OT loss.” Unless it was meant that the model using only OT loss surpassed the model using only MM loss, this statement does not align with what is shown in the table. These sections should be reviewed to ensure the paper is strong in its entirety.

---

> ### Author Response · Authors · 2024-07-08
>
> Thank you for raising several important points regarding our work.
>
> > My biggest concern of this submission is that all the reimplementation details have not been confirmed by the authors of LICO, which makes me difficult to determine if all the results in the experiments are correct and convincing. I have noticed that the authors of the submission discussed the bad communication with the authors of LICO, but this should not be the reason that I have to trust all the results and claims of the submission. Also, from the limitations in the conclusion section, it seems that the reimplementation cannot reproduce some results in LICO paper, such as on ImageNet, and the authors of the submission are not clear about some implementation details of LICO as well. These discussions make all the findings in the submission much less convincing to me.
>
> We have attempted to communicate with the authors to obtain some implementation details, but as we write in the paper, they have not provided the information. It thus is also not feasible to have our entire codebase analyzed by an author of the LICO paper. However, the main purpose of our work was to verify whether it would be possible to reproduce LICO paper based on the publicly available information (paper + codebase). Our experiences during the work on this reproduction show the importance of sufficient documentation of the conducted experiments as the unwillingness of the authors to engage in the discussion should not make it impossible to replicate the results without a lot of otherwise avoidable guesswork. Since the primary claim of our paper is that LICO is not reproducible, we do not see a way to confirm the correctness of our implementation by reproducing all of the original results as we have already seen discrepancies in the experiments conducted by the original authors.
>
> We would like to raise some additional concerns regarding the reproducibility on the ImageNet-1k dataset. First, we observe that the accuracies for ImageNet-1k baseline exactly correspond to the checkpoint provided by PyTorch. However, we also note that many of the reported training hyperparameters are different from the ones used by PyTorch. Specifically: the learning rate is 0.03 instead of 0.1, the learning rate schedule is cosine decay rather than exponential with discrete steps (divide by 10 every 30 epochs), and the batch size is 128 and not 256. The paper also writes that these are the parameters used for all experiments on ImageNet-1k, which must include the baseline.
> No hyperparameter search is mentioned, and it is unlikely that their implementation would achieve the exact same performance that is achieved by the PyTorch baseline, with different hyperparameters. To investigate this, we performed a full run of the PARN18 baseline with both the PyTorch and the LICO hyperparameters. We obtained 0.6956 top1 accuracy & 0.8920 top5 accuracy using the PyTorch parameters, which corresponds to the accuracy reported by PyTorch (and also LICO authors), confirming the correctness of our (baseline) training setup. We obtained 0.5705 top1 accuracy & 0.8090 top5 accuracy using hyperparameters from the LICO paper, which is significantly lower, and indicates that the authors possibly did not even run the baseline using the claimed hyperparameters.
>
> Thank you for pointing out our writing inaccuracies. Claim 3 in section 4.1 has been updated to match the results in Table 2. The claims of the following paragraph were made more clear. A “the” has been added in the SCH section.
>
> > I think that adding the original images used in figure 2. alongside the saliency maps would allow readers to more easily see where the model’s attention is being placed.
>
> Thank you for this suggestion - this point has also been raised by other reviews. We have added the original images to the plot to highlight that LICO saliency maps tend to focus on features outside the boundaries of the classified objects (often random segments close to the image boundary).

---

### Review · Reviewer_Th9w · 2024-05-18

**Summary Of Contributions:**

This paper aims to reproduce the algorithm implemented in another paper called “LICO: Explainable Models with Language-Image COnsistency”. In the absence of fully reproducible open-source code, the authors reimplement the model presented in the original paper. Based on experiments on CIFAR-100 and a subsample of ImageNet-S, this study concludes that the main claims of the original paper could not be reproduced.

More specifically, the original paper presents a new training framework (LICO) that generates saliency maps that would improve both the explainability and the accuracy of image classification models by adding two new components to the Cross Entropy loss function (Manifold Matching and Optimal Transport Loss). The authors of this study couldn’t reproduce the improvement in explainability nor accuracy, and couldn’t reproduce either the effectiveness of the new two components added to the loss function.

**Audience:**

Yes

**Broader Impact Concerns:**

None.

**Claims And Evidence:**

Yes

**Requested Changes:**

Following is specific feedback per page, denoted as p:

## Critical changes

- **Change 1 - p1 in Abstract**: *“The main claims are that LICO…”*. It is confusing whether the claims that are going to be introduced are the ones from this reproducibility study or from the original LICO paper. After reading the paper, it becomes clear that these are claims from the LICO paper, but on a first read this is not obvious. Adding something like *“from the original paper”* or similar could help.

- **Change 2 - p3**: *“incorporates two losses in addition to the standard Cross-Entropy”*. It is not stated how they are incorporated which is important to understand the original paper.

- **Change 3 - p3**: *“The image manifold is defined by feature maps fθ(xi) from the classifier…”*. What classifier? As stated in the weaknesses, this paragraph and the overall section might benefit from clarification.

- **Change 4 - p4**: *“parameters are also used unchanged: α = 10, β = 1”*. Related to change 2, these parameters are presented without context. The final loss function should be presented or at least explained. It is not explained anywhere that the 3 losses, cross entropy - CE, Manifold Matching - MM and Optimal Transport - OT, are added and that α and β are used to control MM and OT.

- **Change 5 - p4**: *“Our code diverges from the formulas by Lei et al. in the way the temperature parameter of the MM loss is implemented. Inspired by the implementation of CLIP (Radford et al., 2021), we multiply by the exponent of the trainable value instead”*. If your code diverges from the original paper, how is the comparison acceptable? If this changes won’t affect the reproducibility, please explain why.

- **Change 6 - p4**: *“share unblurred”*. This expression is confusing. The original paper calls this "pixels removed" which is much clearer. As stated in weaknesses, this whole section where insertion and deletion are explained, could be improved as it is confusing for such simple concepts.

- **Change 7 - p5**: *“The LICO method aims to change the model itself in order to make it more interpretable. Therefore, the comparison between the insertion and deletion scores of a baseline model and a model trained with LICO is not sufficient to claim that one model is more interpretable than the other.”*. This is a strong claim that might need more justification. The justification from Appendix A seems not satisfactory.

- **Change 8 - p6**: *“we select smaller datasets”*. Do you mean smaller batch sizes? If you actually mean datasets, how is the comparison fair?

- **Change 9 - p7**: *“Experiments on ImageNet-S50 using Insertion and Deletion metrics in Table 2 show that training with MM loss achieves a Combined score that is better compared to a ResNet-18 baseline”*. Is this true? In table 3, the best numbers are bolded in the first row, which means that the rest are worse. How is using MM loss better than using ResNet-18, which is the baseline?

- **Change 10 - p7**: *“meaning that the post-hoc explanations produced for LICO models are less robust to image transforms”* How is this conclusion reached?

- **Change 11 - p9**: *“MM loss results in a tiny improvement.”*. Same as Change 9.

- **Change 12 - p12**: *“as it is likely to start using the red dot as the sole predictor of the modified class, rather than the actual object in the picture. We train a ResNet-18 with CE loss from scratch on this dataset and observe that high importance is assigned to the location of the red dot for the class that has it.”*. How is this proven? This point is important because it justifies one of the main claims of the paper: that insertion and deletion metrics are not good enough.
- **Change 13 - p12**: *“no augmentation for the validation and training sets”*. Should it be test sets instead of training?

## Changes that would strengthen the work

- **Change 14 - p4**: *“There is no ReadMe file”*. At the time of this review, there is a README. However, it is still not useful to replicate the model.

- **Change 15 - p5**: *“Where v_salience ∈ Rn denotes”*. *V_salience* could be accompanied by a formula like *v_area*. Also, explain what *i* is, in that case.

- **Change 16 - p7**: *“Figure 2”*. The original images can’t be distinguished in the Figure so I can’t tell if the saliency maps are accurate or not. Either the original images should be shown or the transparency should be higher so that the original image could be distinguished.

**Strengths And Weaknesses:**

## Strengths

1. The authors make a commendable effort in trying to reproduce the work of the original paper, even more considering that the original authors never replied to their inquiries. They create a new repository where they upload both the code and the weights of the models used in the experiments that they conduct. The experiments conducted are not exactly the same as the ones presented in the original paper, due to limitations in hardware (e.g.: they didn’t present results on ImageNet-1k), but they try to make them as comparable as possible (e.g.: presenting ResNet-18 results on CIFAR-100).

2. The additional evaluation metrics used to assess explainability. Namely, they use *Salience Equivariance Similarity (SES)*, Segmentation *Content Heatmap (SCH)* and the *Multi Object Salience Uniformity (MOSU)* in addition to the two metrics used in the original paper: *insertion* and *deletion*.

3. The authors present two very interesting extensions to the way the LICO model uses the text encoder from CLIP. In particular, the authors:
    - Use class-specific context tokens (the original CLIP uses the same tokens for every class).
    - Position the label token at the beginning of the prompt so that it is also encoded (the original CLIP positions the label at the end).

    These two avenues might warrant further exploration on their own.

4. The environmental impact analysis is a very nice touch that more papers should show.

## Weaknesses:

1. The experiments conducted do not completely and exactly replicate the experiments of the original paper. Some of this is unavoidable as the original authors didn’t disclose certain information: e.g. whether to use L2 normalization or not on visual and text features. However, additional experiments could have been conducted. For example:
    - Table 3 should present results for ResNet-50.
    - Also, authors didn’t experiment with ImageNet-1K: while the authors justified it as not having enough computational resources, this might be important given the significance of the claims (that no result could be reproduced).

2. Related to the previous point, the experiments conducted in this study used 16-bit precision. The original paper does not specify the precision used, but if they employed 32 or 64-bit precision, that could explain the difference in results and could discredit this reproducibility study completely.

3. Some parts of the paper are not clear enough and require the reader to consult some references attentively. Specifically, section 3.1 where the LICO algorithm is described and section 3.4.1 where insertion and deletion metrics are presented could benefit from a clearer explanation and would help the paper be self-contained.

---

> ### Author Response · Authors · 2024-07-07
>
> Thank you for the detailed feedback on our reproduction.
>
> We have incorporated the requested changes in the revision of our paper and provided explanation for some of our choices or limitations. In particular,
>
> > Change 1 - 4:
>
> We have slightly expanded on our explanation of the idea of LICO method with an earlier mention of Vision-Language Model used to guide the training process of image classifier and mentioning alpha/beta parameters in context of the overall LICO loss.
>
> > Change 5:
>
> Even given our change, the calculation of the MM loss remains the same for all practical purposes. In fact, we argue that in an implementation where the temperature parameter is learned and unbounded as can be inferred from the LICO paper, the manifold matching loss is unstable around 0 - we have added a stronger emphasis on this point in our revision. Moreover, even ignoring the possibility of division by zero, the loss has a trivial solution by setting the temperature as close to 0 as possible on the negative side, or +infinity on the positive.
>
> We admit that we can not be sure that our implementation of this part (such as the choice for bounds) is fully in line with the authors’. However, as the original implementation is not available, and the paper does not touch on the problem of the trivial solution, we believe this further strengthens our claims about the non-reproducibility of the LICO method.
>
> > Change 6:
>
> We have provided more detailed description of the components used in calculating Insertion/Deletion scores.
>
> > Change 7:
>
> The basis for our claim of Insertion/Deletion metrics inadequacy is the fact that they were designed to assess the quality of saliency maps produced by post-hoc methods obtained by the same model. As their values strongly depend on the classification performance (accuracy) of the tested model, their assessment of quality of saliency maps cannot be trusted when aiming to compare only the interpretability between two or more different models. Moreover, Insertion/Deletion scores can be high even in cases of models displaying abberant behaviour such as overreliance on spurious features. While this is the intended behaviour for post-hoc interpretability methods (as they should attend to the regions actually used by the model), the high values of these metrics when comparing different training procedures are not guaranteed to indicate good properties of the model as understood by attendance to interpretable features of the objects.
>
> Meanwhile, the metrics introduced by us do not depend on model’s predictive performance, meaning that only the saliency maps produced by the post-hoc method for the given model are used to calculated the scores. Additionally, by utilising the ground truth about exact locations of objects in the image, our methods can be used to validate whether the considered models attend to the correct elements of the input rather than spurious features.
>
> > Change 8:
>
> Due to limited computational capabilities, we may not be able to get the results for experiments on the full ImageNet-1k before the intended revision deadline. However, we believe that our original choice of datasets should be sufficient for indicating the problems of reproducing the results of LICO paper - for our revision, we provide our extended justification for our choice of datasets in section 3.2 of our paper, which is a follows: "The selection of these datasets, and not full ImageNet-1k, has been dictated mostly by the computational limitations we have encountered during our work. However, we believe those to be sufficient to show whether the main claims of the LICO paper hold. As the original authors have used CIFAR-100 in their work, we should be able to see the same trends present in our results as they have. However, we acknowledge the fact that limited resolution of images within CIFAR-100 may limit the ability of the method to connect the visual cues with semantic information from VLM. For this reason, we have chosen to conduct additional experiments using ImageNet-S50 dataset, a subset of the original ImageNet-1k limited to 50 classes. The higher resolution with images remove the possibility of LICO's impact being limited by the lack of distinguishable features in blurry images of CIFAR-100. An additional benefit of this datasets is the presence of segmentation masks and bounding boxes for objects, which we utilise in our extended analysis in the Section 4.2.1."
>
> > Change 9, 10, 11, 13, 14, 16:
>
> We fixed the minor mistakes in our writing and provided original figures in the figure 2.
>
> > Change 15:
>
> Thank you for pointing out insufficient explanation on our part. To alleviate it, we provide a similar formula for v_salience and and explanation of index i. We have also provided additional figure demonstrating the idea of all components of our extended evaluation setup to provide intuitive understanding of our methods.

---

### Review · Reviewer_bc35 · 2024-06-24

**Summary Of Contributions:**

The authors aim to test the reproducibility of another work (LICO) which claims to produce a more interpretable image classification model. LICO claims to achieve better interpretability and performance using different training losses to align the image features with the corresponding class labels encoded as text features obtained from a pre-trained vision-language model. The authors of this paper refute the claims of improved explainability and performance on image classification tasks. They also point to the shortcomings of the training approach suggested in LICO and propose improvements to it.

The chief observations in support of their view made by the authors include:
1. the incorrect use of the deletion/insertion metrics in the context of the formulation of LICO. They claim that the deletion/insertion metrics favor non-robust models and cite this as a reason for the inappropriate use in this context. They also suggest using three other metrics - the Salience Equivariance Similarity (SES), the Segmentation Content Heatmap (SCH), and the Multi-object Salience Uniformity (MOSU) to measure the accuracy of explanations. The authors propose the MOSU metric to measure the attribution of the classification result to all instances of the class in the image.
2. the non-reproducibility of the explainability and performance metrics reported in the original paper, for the CIFAR-100/ResNet-18 combination.
3. lack of generalization of the claims to other datasets, notably ImageNet-$S_{50}$ - both quantitatively and qualitatively.
The authors also point to notable omissions such as the normalization scheme used.

In addition to all the results that refute the claims made in LICO, the authors also discuss two simple extensions to improve the salience maps - one through additional learnable parameters by way of class-specific learnable context tokens (as opposed to identical trainable tokens used in LICO), and second, by placing the class label in front of the context tokens so as to influence the latters' encoding through the causal attention mechanics which the encoding transformer typically employs.

**Audience:**

Yes

**Broader Impact Concerns:**

The paper could be a bit less adversarial (for example, in pointing out omissions). This would encourage the original papers' authors to approach the critique more constructively.

**Claims And Evidence:**

Yes

**Requested Changes:**

The authors are requested to revisit their premise for the rejection of the deletion/insertion score methods, as well as to state their reasons for testing their position only for ResNet-18. They are also requested to furnish evidence of the efficacy of the alternative metrics for measuring the salience map explainability.

**Strengths And Weaknesses:**

Strengths:
* The paper identifies some major issues with the reproducibility of results. Albeit, it must be pointed out that the implementation by the authors differs in small ways (for instance, the temperature parameter of the Manifold matching loss). In particular, the non-reproducibility of the deletion/insertion scores for the ResNet-18 + CIFAR-100 combination is a key indicator that the LICO method may not be universally applicable. It is noteworthy that the original LICO paper does not disclose the top-1/top-5 accuracy scores for ResNet18+CIFAR-100 at all.
* The ImageNet-$S_{50}$ result is also telling, indicating the lack of generalizability of the LICO approach to a different architecture + dataset combination.
* The paper is well-written and points to some important considerations for the measurement of saliency map efficacy for explanations.

Weaknesses:
There are at least three important objections to the conclusions stated in this paper:
1. The non-applicability of the deletion/insertion score on the pretext that it is applied only to a model+salience-map-algo pair when the model is frozen (whereas, in LICO, the visual encoder is trained), and that the metric favors small regions in the image.  Both assertions may be untrue. In the first case, while the LICO encoder is indeed trained to produce better saliency maps, it is not explicitly trained to do so and so, any post-hoc explanation mechanism and metric may be employed at inference time. Secondly, the authors provide evidence for the latter assertion through what they term an adversarial training dataset where *all* images of a class are augmented with a red dot and the saliency maps indicate as such. The catch here is that this is exactly the expected behavior for any saliency map generation method and metric. Instead, for example, if the model was actually adversarially attacked via a low-\epsilon attack and then the salience map was found to favor the adversarial regions, it may have indicated an incorrect bias in favor of non-robust models.
2. All the stated results are given for ResNet-18, and not ResNet-50 which is the main network used by LICO for most of its experiments. While this does not absolve the original paper from signaling an incomplete conclusion, the results may still be valid for models larger than a certain size. A paper refuting its claims must test LICO with both smaller and larger networks.
3. The proposed alternative metrics, while intuitively appropriate, do not have backing evidence. For instance, the premise of the MOSU metric is that to classify an image with multiple instances of the target class, each of those instances must be attributed is an unfounded one with no supplied evidence (citations, etc.) for itself. The paper is also missing the element of human judgment for the efficacy of these new metrics.
In addition, the premise that the attribution for a particular classification must be limited to the boundaries of the object is not substantiated with empirical evidence or prior art. In some cases, the classification may be inferred from the surrounding context, especially if significant portions of the object in question are occluded.

---

> ### Author Response · Authors · 2024-07-07
>
> Thank you for constructive feedback on our reproduction effort.
>
> > The non-applicability of the deletion/insertion score on the pretext that it is applied only to a model+salience-map-algo pair when the model is frozen (whereas, in LICO, the visual encoder is trained), and that the metric favors small regions in the image. Both assertions may be untrue. In the first case, while the LICO encoder is indeed trained to produce better saliency maps, it is not explicitly trained to do so and so, any post-hoc explanation mechanism and metric may be employed at inference time. Secondly, the authors provide evidence for the latter assertion through what they term an adversarial training dataset where all images of a class are augmented with a red dot and the saliency maps indicate as such. The catch here is that this is exactly the expected behavior for any saliency map generation method and metric. Instead, for example, if the model was actually adversarially attacked via a low-\epsilon attack and then the salience map was found to favor the adversarial regions, it may have indicated an incorrect bias in favor of non-robust models.
>
> Our first assertion is not quite that the model has to be frozen rather than trained. Rather, we point out that insertion and deletion metrics were proposed to assess saliency maps produced by post-hoc methods for the same model. One could take a specific model, and apply multiple methods for saliency maps extraction, such as Grad-CAM or RISE. Then, using the insertion and deletion score, it is possible to argue which of the saliency extraction methods performed better in explaining the particular model. On the other hand, as we discussed, it does not make sense to compare different models by how much they score in insertion and deletion as it is difficult to properly define the meaning of model's "interpretability" in this context - the model may be easier to interpret a post-hoc on the testing images, but if it is achieved by a model being non-robust and focusing strongly on only a single, potentially spurious feature, it is generally not an intended behaviour. As LICO changes the model, the two metrics are not applicable to assessing the success of the LICO method in increasing the "interpretability" of the trained model.
>
> LICO is designed to improve the saliency maps using the help of a pretrained model - CLIP. Thus, it may be expected that thanks to that model's knowledge and the additional loss components, even a high epsilon attack can be circumvented. Our experiment shows that it can not be circumvented, and the model trained to be non-robust, "outperforms" the robust model on the metrics suggested by the LICO authors.
>
> > All the stated results are given for ResNet-18, and not ResNet-50 which is the main network used by LICO for most of its experiments. While this does not absolve the original paper from signaling an incomplete conclusion, the results may still be valid for models larger than a certain size. A paper refuting its claims must test LICO with both smaller and larger networks.
>
> While this is a consideration we had in mind when planning our original experiments, in our analysis of the LICO paper we did not find any reasoning or indication about results being only applicable to larger networks. Furthermore, many of the experiments of the authors'  have been conducted also on PARN18, which is what we are using for all of our experiments due to computational constraints and the trends visible in those have not been observed in our work. We believe this to be a strong indication at non-reproducibility of the original paper.
>
> > The proposed alternative metrics, while intuitively appropriate, do not have backing evidence. For instance, the premise of the MOSU metric is that to classify an image with multiple instances of the target class, each of those instances must be attributed is an unfounded one with no supplied evidence (citations, etc.) for itself. The paper is also missing the element of human judgment for the efficacy of these new metrics. In addition, the premise that the attribution for a particular classification must be limited to the boundaries of the object is not substantiated with empirical evidence or prior art. In some cases, the classification may be inferred from the surrounding context, especially if significant portions of the object in question are occluded.
>
> The primary reason to introduce MOSU as one of the quantitative evaluations in our paper is the claim by LICO authors: “However, the baseline methods are inferior in capturing all the objects in the multi-object image.” in the qualitative results section. As there are no numbers to support that claim, we decided to make a metric that would measure this. Our experiments show that MOSU score is worse for LICO than it is for the baseline. On the other hand, SES and SCH are both based on prior works - with SCH being an extension of Content Heatmap metric used in CGC paper to segmantation masks.

---

### Decision · Action_Editor_WDqP · 2024-09-09

**Recommendation:** Accept with minor revision

**Comment:**

Taking the stance that we are primarily evaluating correctness, this paper should be accepted (with slightly softened language). However, if were evaluating "impact", then I would give this paper a reject. The original work is just not of sufficiently well known to expect broad audience for a paper that attempts a reproduction.

**Audience:**

Yes, potentially. I imagine that it could be considered a reasonable example of this sort of a reproduction paper. However, I don't anticipate that too many people will read this paper because they tried to implement the LICO paper.

**Claims And Evidence:**

This paper is a little untypical as this is an attempt to reproduce another paper (LICO) rather than original research. If we assume that the claim is "the LICO paper can't be reproduced" using the information provided in the paper and the associated repository, then this paper is probably executed correctly. At least I have not identified anything to be convinced otherwise. There is clear evidence for at least that it is very difficult to reproduce the original paper, if the results are even correct. The authors put significant effort into attempting to execute that. Having said that, I think realistically this paper is of very low impact (the original LICO paper has been cited only 4 times so far). Furthermore, the paper is a little bit too verbally adversarial to my taste. I understand the frustration of not being able to reproduce a paper published at a major conference, however, some things could be phrased more politely (please make an honest attempt to do that, as well taking all points raised by the reviewers in their final evaluations).